# Cardiovascular Diseases, Vital Organ Fibrosis, and Chronic Inflammation Associated with High-Intensity and/or High-Volume Exercise Training: Double-Edged Sword Effects of Vigorous Physical Activity in Elderly People and/or in Middle-Age Cancer-Therapy-Treated Patients

**DOI:** 10.3390/jfmk10010033

**Published:** 2025-01-15

**Authors:** Pierre A. Guertin

**Affiliations:** Department of Neurosciences, Faculty of Medicine, Laval University, Québec, QC G1V 0A6, Canada; pierre-a.guertin@fmed.ulaval.ca

**Keywords:** cardiotoxicity, hepatotoxicity, lung toxicity, marathon, ultrarace, scar tissue, HIT, aging, chemotherapy, radiotherapy

## Abstract

Cardiotoxicity, cardiovascular diseases (CVDs), hypertension, hepatotoxicity, and respiratory problems occurring several months to several years post-chemotherapy and/or radiotherapy are increasingly documented by scientists and clinicians. Anthracyclines, for example, were discovered in the late 1960s to be dose-dependently linked to induced cardiotoxicity, which frequently resulted in cardiomyopathy and heart failure. Most of those changes have also been associated with aging. While it is well known that exercise can slow down cellular aging processes, lessen the effects of chemotherapy, improve the effectiveness of cancer treatments, and prevent health problems in the general population, it remains unclear how exercise volume or intensity may affect the overall benefits of physical activity on health. For instance, higher rates of sudden cardiac arrest or coronary artery calcification have been found in marathon and ultra-marathon runners. Several additional pathological consequences have also been reported recently on many organs of those athletes. This review reports the most recent evidence suggesting that excessive intensity and/or volume may have deleterious effects on health. These findings are in clear contrast with the popular belief that all forms of physical activity can generally reduce the pathological changes associated with aging or cancer therapies. In conclusion, high-intensity training (HIT) and/or high-volume training (HVT) should not be recommended for middle-age and elderly people who have had cancer therapies or not in order to avoid an exacerbation of the consequences of aging or long-term cancer treatment effects on vital organ structures and functions.

## 1. Introduction

It is well known that most cancer treatments can have side effects affecting healthy tissues and organs—e.g., hair loss, neutropenia, fatigue, etc. However, fewer people know that a significant list of long-term consequences on health are also found in survivors. These include long-term cancer-related fatigue (CRF), bowel or bladder problems, sleeping problems, mental distress, changes in memory and attention, hearing problems, osteoporosis, endocrine disorders, low androgen levels, sexual dysfunction and fertility problems, cardiotoxicity, cardiomyopathy, cardiovascular diseases (CVDs), secondary cancers, liver diseases, etc. Interestingly, most of those problems are also experienced naturally by the elderly over time. Whether or not all forms of exercise training can prevent these pathophysiological changes remains unclear.

## 2. Long-Term Effects of Cancer Treatments

With almost nine million fatalities per year, cancer ranks among the world’s top causes of mortality. The World Health Organization (WHO) projects a nearly 70% increase in cancer incidence over the next 20 years [1]. It is therefore reasonable to predict a growth in the projected number of cancer survivors who develop life-threatening conditions and other issues as a result of side effects from radiation therapy or cancer drugs. Cardiotoxicity is a well-recognized adverse consequence of radiation or chemotherapy that dramatically increases rates of morbidity and death. Cardiotoxicity can appear at any stage of the disease’s development, from moderate cardiac dysfunction to irreversible heart failure leading to death [2]. Cancer survivors have 15-fold greater risks of experiencing long-term life-threatening consequences such as congestive heart failure.

More and more scientists discover that most cancer treatments have an impact on nearly all vital organs. For example, it was recently found that the range of pulmonary toxicity related to chemotherapy was between 0.1 and 15% [3]. Inflammation and fibrosis may arise from lung damage, sometimes several years after chemotherapy is administered. Bleomycin, a chemotherapeutic drug with potent anti-tumor properties, has been specifically associated with a high incidence of pulmonary problems [4]. The pulmonary inflammation and fibrosis induced by that drug are aggravated by oxygen administration or radiotherapy (RT). Other anti-cancer agents associated with significant long-term pulmonary damage include carmustine, methotrexate, busulfan, and mitomycin [5].

The liver is also severely impacted by cancer therapies. Under chemotherapy, up to 85% of patients develop liver problems such as steatosis and over time, steatohepatitis, a more serious long-term consequence [6]. For example, fluorouracil has clearly been shown to lead to the development of hepatic steatosis [7].

Other vital organs such as the pancreas and biliary system are also particularly impacted by chemotherapy [8]. Ultimately, the long-term effects of cancer therapies might vary greatly depending on the type of treatment (e.g., chemotherapy vs. radiotherapy), the particular form of cancer, and the patient’s overall health prior to treatment. Chemotherapy is generally associated with long-term, sustained fatigue and damage to the heart, lungs, kidneys, and liver; it also raises the risk of peripheral neuropathy and secondary cancers resulting from DNA damage (e.g., acute myeloid leukemia); on the other hand, radiation therapy is more closely linked to long-term damage (e.g., fibrosis) and secondary cancers in tissues and organs adjacent to the treated area. Additionally, it has been discovered that immunological and hormonal therapies have particular long-term negative effects on the control of hormone release and immune function (e.g., hormonal therapies may increase the risk of osteoporosis and fractures) [9,10] (Figure 1).

Chemotherapeutic drugs mainly responsible for these long-term complications include anthracyclines (e.g., doxorubicin, epirubicin, daunorubicin, idarubicin, cyclines) as mentioned previously but also antibiotics derived from Streptomyces that have been widely used for more than 50 years against leukemia, lymphoma, and breast, stomach, uterine, ovarian, bladder, and lung cancers [11,12].

**Figure 1 jfmk-10-00033-f001:**
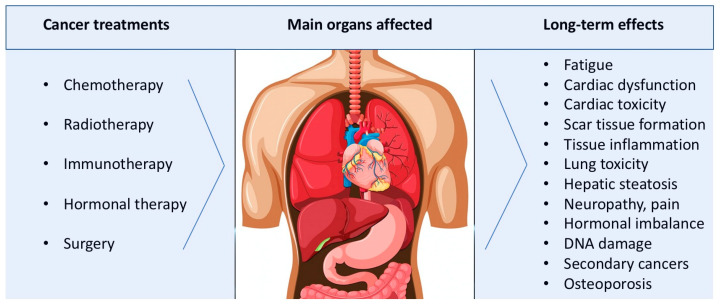
Main long-term/lifetime effects of cancer treatments on health and vital organ functions. Part of this royalty-free image designed by brgfx is from Freepik. For further details on specific treatments and their corresponding long-term complications, see Thong MSY et al. [13].

## 3. Natural Consequences of Aging on Systemic Functions and Vital Organs

According to a recent report from the United Nations (UN) in 2023, there were 761 million individuals aged 65 years or older in 2021. The number of elderly people is growing rapidly. Population aging is an irreversible global trend—the number of people aged 65 years or older worldwide is indeed projected to more than double, rising from 761 million in 2021 to 1.6 billion in 2050 [14]. Everyone is well aware that aging has an impact on health. Generally, pathological conditions associated with aging include hearing loss, cataracts, chronic pain (e.g., lower back pain), osteoarthritis, chronic obstructive pulmonary disease (COPD), mobility and equilibrium problems, reduced reflexes, diabetes, depression, and dementia. Beyond those expected health concerns, it turns out that aging causes more or less the same changes to vital organs as radiation or chemotherapy when examined at the cellular level.

Aging decreases heart rate variability and increases autonomic nervous system imbalance, cardiac cell hypertrophy (e.g., cardiomyopathy), cardiac interstitial fibrosis, arterial stiffness, and mitral valve dysfunction [15,16,17,18]. Aging alters also hepatic cells and liver fibrosis [19]. The probability of pancreatic fibrosis in people aged 60 years and older is 62.0%—aging is indeed associated with changes in volume, dimension, contour, and increasing intrapancreatic fat deposition in the pancreas [20]. These changes involving the pancreas lead to a decrease in perfusion, fibrosis, and atrophy. Most skeletal muscles are also significantly affected by aging. Sarcopenia is a steady decrease in muscular mass and strength that occurs with aging. On average, muscle mass decreases by 3–8% every decade after the age of 30—a higher rate of muscle tissue loss is naturally found after the age of 60 [21]. Beyond muscle problems, although a link between muscle tissue loss and bone loss exists, decreased bone density and increased risks of fracture are hallmarks of aging. Aging-related bone loss is due to imbalanced bone resorption relative to bone formation. Specifically, a shift of increased osteoclasts from trabecular to cortical bones is normally found in elderly people [22]. As aging increases, the kidney undergoes also significant changes—e.g., reduced number and size of nephrons, glomerulosclerosis, tubular atrophy, inflammation, interstitial fibrosis, etc. [23,24] (Figure 2).

## 4. The Established General Benefits of Exercise Training on Health

Physical activity and specifically aerobic exercise training are generally known to improve the cardiovascular system and prevent the development of corresponding diseases such as metabolic disorders, obesity, type 2 diabetes, and more https://www.cdc.gov/physical-activity-basics/benefits/index.html (accessed on 12 January 2025)). As such, it has been typically recommended for decades by medical doctors and cardiologists for everyone—and most particularly for individuals at risk of developing CVDs and vital organ dysfunctions such as elderly people or for those who have had cancer therapy (middle-age or older). It is also recommended to prevent respiratory problems. Exercising regularly can enhance lung capacity and improve the efficiency of the respiratory system given that the lungs work harder to supply oxygen to the body, which strengthens the respiratory muscles and improves their function. This increased efficiency helps prevent respiratory conditions such as COPD.

Unsurprisingly, physical inactivity, paralysis, and a sedentary lifestyle have been associated with opposite effects—they induce pro-aging (accelerated aging) effects on the body—e.g., walking or being moderately active physically less than 30 min per day generally increases the risks of developing obesity, infections, osteoporosis, metabolic disorders, cardiovascular problems, and dyslipidemia in able-bodied persons. Walking less than 30 min per day has been associated with an increased incidence of insulin resistance, hypertension, cholesterol, incidence of type II diabetes, and other cardiovascular problems [25]. However, although exercising regularly is generally good for health, the role of volume and intensity of training on vital organ physiology remains unclear [26,27].

## 5. Risks of Excessive Physical Activity on Organ Structures and Functions

Until a few years ago, the general public believed that long-distance runners, such as marathoners, were immune to respiratory and cardiovascular issues because their risk of cardiac mortality was considered to be relatively low [28]. However, increasing evidence suggests that discrepancies exist. It is indeed not unexpected for long-distance runners to experience sudden cardiac death (SCD), musculoskeletal injuries, gastrointestinal problems, and life-threatening hyponatremia due to overhydration with hypotonic fluids [28,29]. SCD is a non-violent, non-traumatic, and usually unanticipated event that happens after or following tremendous efforts. In fact, even runners without a history of cardiac issues can experience cardiac arrest or ventricular arrhythmia after engaging in prolonged activity. Instances of SCD in regular (or “healthy”) long-distance runners can occur at a ratio of 1 in 15,000 [30,31]. When compared to the general population (i.e., persons who do not participate in high-volume, long-distance sports), the incidence among marathoners nearly doubles during competition, particularly in the final kilometers [32].

Researchers from Spain have presented data that could indicate that running a full marathon is especially taxing on the heart. Compared to athletes competing in half-marathons or 10 K runs, marathon runners exhibited higher levels of biomarkers of aberrant cardiovascular stress, such as myoglobin concentration, muscular soreness, body mass loss, and electrolyte imbalances [33].

Other studies reveal that blood analyses performed less than 24 h after finishing a long-distance race exhibit abnormally high levels of inflammatory and clotting markers such as creatine kinase-MB and B-type natriuretic peptide, typically recognized to be related to heart attacks [34].

Dr. McCullough, Chief of cardiovascular research at the Baylor Heart and Vascular Institute and a man who has completed 54 marathons in his lifetime (and who no longer runs given his breakthrough findings on this matter), pointed out that during three hours or more of high-volume cardiac overload such as during long-distance races, the right atrium and right ventricle undergo abnormal dilatation which, in turn, is associated with temporary impaired chambers attributed to blood troponins and B-type natriuretic peptide increase [35]. “Our theory is that 25 percent of people are susceptible to this recurrent injury of the heart”, McCullough said. “A smaller subset, about 1 percent, could be prone to myocardial fibrosis or scarring of the heart, that can lead to heart failure”, according to him.

After a 140-day cross-country race, an increase in c-reactive protein (CRP) levels—a biomarker of inflammation—and calcium plaque build-up in the arteries have been found in runners during the race [36]. According to the authors of that study, extreme long-distance running may not be protective against heart disease in the same way that regular exercise is. Their findings suggest instead that extreme exercise may actually accelerate the progression of heart disease. This is also supported by another study published in 2019 reporting significantly higher rates of coronary artery calcification in long-term marathon, ultra-marathon, and extreme runners compared with sub-marathon runners [37].

Overuse injuries associated with repeated stress and unresolved inflammatory mechanisms are also often found. These include patellofemoral pain syndrome, iliotibial band friction syndrome, medial tibial stress syndrome, Achilles tendinopathy, plantar fasciitis, and lower-extremity stress fractures [38]. Approximately 21% of female endurance runners develop stress fractures attributed to increased energy expenditure concomitant with inadequate nutrition [39]. Moreover, endurance athletes are more at risk for exercise-associated asthma, collapse, overtraining syndrome, immunodepression, and exaggerated inflammatory response affecting the upper respiratory tract (e.g., two-fold increase in bronchoconstriction) and the gastrointestinal system [40,41,42]. During most long-distance events, the distal airways are exposed to unconditioned air that cools and dehydrates the epithelial surfaces resulting in inflammation that can stimulate bronchial smooth muscle constriction, airway narrowing, and subsequent obstruction. When repeated, this may cause injury and remodeling problems of the bronchial smooth muscle.

The renal, gastrointestinal (GI), immune, and neurological systems may also be impaired (Figure 3)—the risk of kidney injury could be exacerbated by factors in extreme environments (e.g., hot and/or humid conditions, racing at altitude), severe muscle damage due to high biomechanical loads, low rates of fluid intake resulting in dehydration, the ingestion of nonsteroidal anti-inflammatory drugs (NSAIDs), and genetic predisposition [43]. Acute GI problems are commonly found during long-distance training and racing given that 50–80% of runners experience nausea, vomiting, and/or diarrhea [44]. Just after a long-distance race, there is a transient immunosuppression (e.g., reduced immunoglobulin IgA levels) for several hours, which normally increases the risk of acute subclinical and clinical viral and bacterial infections—pro-inflammatory factors such as G-CSF, IL-10, IL-6, IL-8, cortisol, and low testosterone are also found [45,46,47]. In general, it is known that strenuous acute exercise can cause immunosuppression, referred to as “the open window theory” or “the elite athlete’s paradox”, which can increase susceptibility to infection [48].

Although MRI data show a 6% reduction in brain volume after ultra-racing and a return to baseline 8 months later, the possibility of long-term detrimental effects has not been systematically investigated yet [49]. That said, compared to other athletes or the general public, ultra-endurance athletes frequently show noticeably higher levels of exercise dependence, which may in theory have a detrimental long-term effect on mental health. (e.g., depression, anxiety, etc.) [50].

## 6. Discussion

Scientists have clearly shown in recent years that chemotherapy and other cancer treatments have long-term (irreversible or not) deleterious effects on health. The latter include tissue damage because chemotherapy agents, designed to target rapidly dividing cancer cells, affect also healthy cells, especially in tissues such as the bone marrow, the GI tract, and hair follicles, leading to long-term complications including anemia, digestive issues, and hair loss. They elicit also reversible or irreversible organ toxicity problems—at least some chemotherapy drugs can cause long-lasting damage to vital organs, such as the heart, lungs, and kidneys. This can result in chronic conditions that may require ongoing management. Certain chemotherapy treatments can lead to long-lasting and often irreversible peripheral neuropathy, characterized by pain, tingling, or numbness in the extremities. Secondary cancer risks are also increased later in life due to the mutagenic effects (DNA damage) of these drugs. Cognitive impairment may also be found since some patients can experience “chemo brain”, including cognitive deficits, memory problems, and difficulty concentrating that can last for years after treatment. In brief, while chemotherapy is a critical component of cancer treatment, its potential for causing long-term, irreversible health issues underscores the importance of careful management and monitoring during and after treatment.

Treatments such as chemotherapy and radiation can lead to a wide range of side effects as mentioned above—e.g., weakened immune function, fatigue, and damage to healthy tissues. These side effects can compromise overall health and quality of life. When getting older, comparable consequences on health and organ functions are also generally found—e.g., physiological changes, increased vulnerability to diseases, reduced regenerative capacity, and declined physical and cognitive functions. Moreover, while regular exercise is generally considered beneficial to health, excessive training can lead to overuse injuries, hormonal imbalances, and increased stress on the body. This can result in fatigue, decreased performance, and a higher risk of illness. In brief, while cancer therapy is necessary for treatment and aging processes affecting most organs of our body are inevitable, it appears reasonable for people concerned to be careful not to get engaged in long-distance performance or strenuous sports because organs already damaged by aging and/or cancer treatment could eventually be further damaged over time.

The detailed cellular, intracellular, and genetic mechanisms underlying aging, long-term cancer treatment side effects, and HIT/HVT deleterious consequences on health are not completely understood. More research is needed to fully comprehend if similar pathophysiological mechanisms and pathways are commonly used or shared by aging, cancer treatment, and excessive physical activity. If that were to be the case, combining some of these risk factors (i.e., previous chemotherapy, aging, and HIT/HVT) could accelerate or augment the risks of life-threatening organ dysfunction or failure. As of now, clear evidence of additive or synergistic deleterious effects remains lacking. Nonetheless, it may appear prudent in those circumstances and given the existing body of evidence to avoid training for marathons, ultra-marathons, triathlons, ultra-triathlons, or other long-distance sports at least for people at risk.

The question is, who are the people at risk? Is it only elderly people and middle-age adults suffering from stress (e.g., professional or familial) or also kids, teenagers, young adults, menopausal women, etc.? As of now, no clear answer exists. That said, aging-related symptoms and diminished organ function start earlier than generally believed in the population. For example, it is commonly known among endocrinologists that testosterone, the hormone that epitomizes youth and energy in males, starts to decline around the age of thirty. Some men in their 20s experience the first symptoms of hair loss. Cognitive capacities also begin to decline at the ages of thirty or forty. A significant acceleration of aging is found in women in their 40s or 50s experiencing perimenopause and menopause (i.e., sudden decrease in estrogen levels—the female’s hormone of vitality, reproductive health, and strength) and in men going through the corresponding phase (male menopause—also known as andropause). For instance, cognitive deficits have been described in women during the menopausal transition, particularly in cognitive domains such as working memory, attention, reduced processing speed, and reduced verbal memory [51]. Menopause has been clearly associated also with an acceleration of bone loss, muscle atrophy, and heart problems. Younger people may also be at risk of practicing long-distance sports for different reasons—there is no consensus on this issue, but it is not unreasonable to believe that bone formation, metabolism, and anabolic hormone release may be compromised by the overall stress on the body induced by excessive physical activity.

Beyond the question of who is vulnerable and who should not practice such levels of physical activity, perhaps the most important question is why. Why do people feel the need to run, swim, cycle, or ski long-distance events? According to statistics reported by the magazine *RunRepeat*, participation in ultrarunning events has been on the rise in recent years. There has been a 1676% increase in participation since 1996. There were just 34,401 ultra-running participations back in 1996, and now there are 611,098 (https://runrepeat.com/state-of-ultra-running (accessed on 12 January 2025)). Several reasons for this trend have been proposed. This increase in ultra-marathon participation has been associated with a rise in the number of ultra-marathon events, athletes’ “migration” from marathons to ultra-marathons, and an increase in the participation of female runners, as well as of both younger and older runners [52]. However, lying underneath are the fundamental motivation and psychological reasons for this trend—some say that long-distance racing keeps the voices in their head quiet (i.e., a search for mental peace), stimulates antidepressant effects (search for better feel-good moods such as dopamine- and endorphin-induced joyfulness), and enables the impression of being alone, relaxation, and creative thinking. Others declare a need to explore their mental and physical limits or to gain certainties about their capacity to reach exceptional objectives. Their search for social interactions (e.g., during racing events) or social approval (i.e., trying to impress their family, friends, colleagues, etc.) can’t be excluded in some cases. Having a relatively simple method to control fitness and body weight is most probably also one of the main reasons for long-distance athletes or amateur runners.

Since several other methods exist to fulfill the needs mentioned above (e.g., having a healthy diet, exploring with a psychotherapist the profound social reasons for performing long-distance sports, undertaking self-reflection/introspection activities to rationally analyze the pros and cons of this particular type of activity known to have some health benefits but also significant deleterious consequences on organ structures and functions), it could make sense to investigate them first before participating in this kind of sport on a regular basis. Understanding fully the reasons why performing long-distance activities may help make a rational decision about whether or not a person should begin, continue, or stop training for and doing marathons, ultra-marathons, or triathlons.

All in all, strong scientific research indicates that increased activity is always preferable and supports the special therapeutic benefits of exercise for lifespan, cardiovascular health, and quality of life. Chronically engaging in excessive endurance training, particularly in ultra-endurance races that carry a risk of myocardial damage, abrupt cardiac arrest, and systemic inflammatory disorders, may have a negative effect on one’s health. While the ideal amount of exercise is still unknown, research indicates that doing 2.5 to 5 h per week of moderate to intense physical activity will have the greatest positive effects on health, while doing 10 h or more may have the opposite effect.

## 7. Concluding Remarks

Given the mounting evidence of serious side effects occasionally caused by HIT or HVT, it may be appropriate to advise middle-aged or elderly people who have received cancer therapies or not, to avoid participating in these activities if exercise in certain situations may be detrimental to health and permanently change the anatomical or functional characteristics of some organs. It is now fair to believe that excessive physical activity could harm the same organs that have already been acutely or irrevocably altered by aging and/or cancer therapy, according to current research. More problems with the heart, liver, pancreas, bones, muscles, brain, or kidneys would most likely result from this.

This being said and as clearly pointed out by the CDC, it remains true that sedentarism increases the incidence of anxiety, depression, metabolic disorders, and obesity (www.cdc.gov/physicalactivity/basics/adults/index.htm (accessed on 12 January 2025)), to name a few. In turn, compelling evidence continues to show that active walking at least 30 min per day several times per week or light and moderate-intensity exercise can reduce the risk of metabolic disorders, CVDs, and stress while improving vital organ functions, sleeping, cognition, gut microbiome activity, immune reaction, and hormonal regulation [53,54,55,56,57]. Globally speaking, maintaining a holistically healthy lifestyle—that is, abstaining from smoking, drinking in moderation, and eating a well-balanced diet—is essential to reaping the full health advantages of frequent, “normal” exercise.

## Figures and Tables

**Figure 2 jfmk-10-00033-f002:**
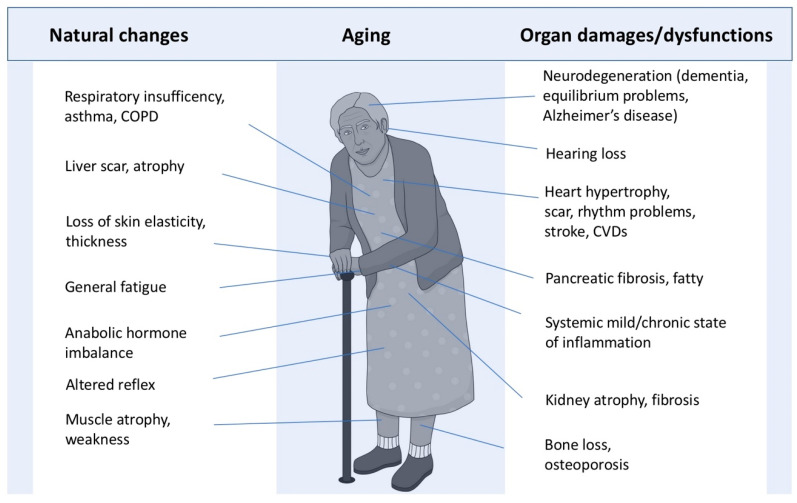
Natural consequences of aging on structure and function in most organs of the body. Part of this royalty-free image designed by grfxrf is from Freepik. For further details about the physiological changes of aging, see Preston and Biddell [24].

**Figure 3 jfmk-10-00033-f003:**
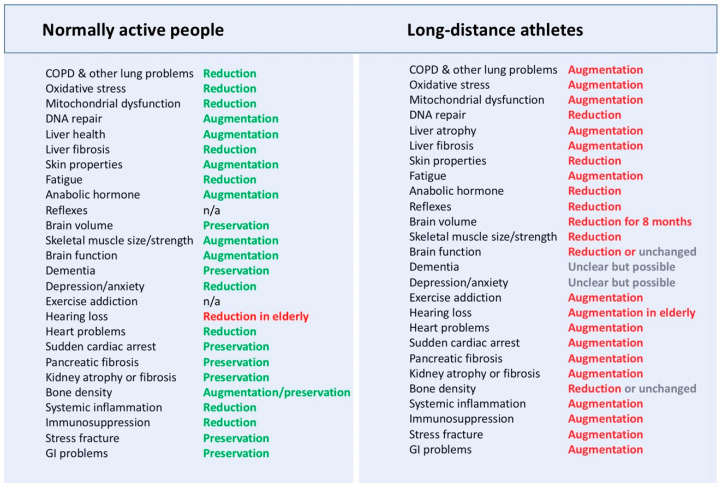
Short- and long-term health consequences of normal vs. long-distance exercising. Text in green and red is associated with positive and negative changes, corresponding.

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
