# Peer review of "Cardiovascular Diseases, Vital Organ Fibrosis, and Chronic Inflammation Associated with High-Intensity and/or High-Volume Exercise Training: Double-Edged Sword Effects of Vigorous Physical Activity in Elderly People and/or in Middle-Age Cancer-Therapy-Treated Patients"

_jfmk, 2025, doi:10.3390/jfmk10010033_

Round 1

Reviewer 1 Report

Comments and Suggestions for Authors

The  paper  approaches  an  important aspect of  the correct  regimen of  the exercise volume  . The intensity  of exercise  is  in fact fundamental to support the subjects   with metabolic chronic diseases , especially those under exercise prescription program .

 The review should highlights aspect regarding  the antioxidant effect of the  exercise , if practiced  at moderate intensity  and clarify  the  role  of pro-inflammatory effect  of  the high volume  of exercise  like in sports  activity .

A definition and a reference formula  to determine a tailored exercise prescription needs  to  be written .

In this term,  the role  of  the antioxidant capacity of moderate exercise should be underlined as mechanism to control  the  cardiotoxicity  due  to  the long time  of exposition to the chemotherapy .

 The  same  mechanisms should be   discussed for the sarcopenia risk in cancer patients .

Can literature supports a different behaviour in case of metastatic or not metastatic cancer  disease?

Author Response

Comment 1. The review should highlights aspect regarding the antioxidant effect of the exercise , if practiced at moderate intensity and clarify the role of pro-inflammatory effect of the high volume of exercise like in sports activity.

Answer. This is an interesting question but no correction will be made. Although there is abundant literature on endurance training and reduced pro-inflammatory cytokine activation, specific links between very high-volume or  high-intensity training and  pro-inflammatory mechanisms underlying pathological conditions remain largely unexplored. I believe that attempting to establish such a link would be hazardous and misleading based on data from studies in animals or humans that have focused simply on 'normal' or 'broadly defined' endurance training. In this case, the review is one of the firsts discussing the existence of preliminary evidence of tissue damage caused by excessive physical activity. The specific detailed cellular mechanisms may have to await further investigations.

Comment 2. A definition and a reference formula to determine a tailored exercise prescription needs to be written.

Answer. No change will be made. This review aimed at uncovering the existence of tissue damage associated with excessive exercise training. It is not aimed to propose specifically what the ideal exercise regimen should be. I only refer to the guidelines provided by the authorities (e.g., CDC).

Comment 3. In this term, the role of the antioxidant capacity of moderate exercise should be underlined as mechanism to control the cardiotoxicity due to the long time of exposition to the chemotherapy.

Answer. No change will be made. Again, this review is not at all about the benefits and mechanisms underlying moderate exercise training. The is numerous existing reviews about that topic. 

Comment 4. The same mechanisms should be  discussed for the sarcopenia risk in cancer patients.

Answer. Interesting topic indeed. However, no change will be made because it is far beyond the matter specifically discussed in this review.

Comment 5. Can literature supports a different behaviour in case of metastatic or not metastatic cancer  disease?

Answer. No, not to my knowledge.

Reviewer 2 Report

Comments and Suggestions for Authors

The manuscript entitled “Cardiovascular diseases, vital organ fibrosis, and chronic inflammation associated with high-intensity and/or high-volume exercise training: Double-edged sword effects of vigorous physical activity in elderly people and/or in middle-age cancer therapy-treated patients?” has been evaluated and presents an important contribution to the literature. However, for it to be published, some adjustments are needed. Below are the necessary demands that I consider necessary:

Topic 2: Long-term effects of cancer treatments

1) Cite the reference used to prepare the paragraph located between lines 79 and 83. Also remove the underlined information in this paragraph.

2) In the caption for figure 1, state where the information was extracted for its creation, as well as explaining better in the text all types of treatment and chronic side effects on the affected organs and/or regions of the body.

Topic 3: Natural consequences of aging on systemic functions and vital organs

3) In line 100, it is mentioned that the age factor is related to the increase in heart rate variability, and is subsequently pointed out as something bad, however, the literature indicates that the greater the heart rate variability, the better the health. Please check this statement and adjust.

4) Regarding figure 2, as mentioned in figure 1, please cite the authors who were used as support for the creation of this image. In addition, detail in the text the possible physiological mechanisms that lead to the outcomes of the changes proposed in figure 2.

Topic 4: The established general benefits of exercise training on health

5) Cite the reference used to create the first paragraph of this topic (between lines 123 and 133).

6) In topic 4, the first paragraph provides information about the importance of physical activity and the second paragraph mentions the harmful effects of physical inactivity. However, this information is very superficial and requires more details, such as mentioning the intensity of the effort (light or moderate), immunological and behavioral changes, body composition and recommendations from the World Health Organization and the American College of Sports Medicine, for example. Here are some supporting articles to strengthen this topic: https://doi.org/10.1186/s13195-020-00597-3; https://doi.org/10.1139/apnm-2020-0467; https://doi.org/10.1590/1517-869220182405185533; https://doi.org/10.3390/medsci8010011; https://doi.org/10.1155/2018/5230971.

Topic 5: Risks of excessive physical activity on organ structures and functions

7) The assertion between lines 194 and 198 needs to be referenced.

8) In figure 3, it is important to mention in front of the topics which author is related to each statement, this makes it easier for the reader to search for the information, if necessary.

9) Before the discussion and after topic 5, I believe it is necessary to create a topic focused on the central theme of the review, which would specifically address the objective of the study, that is, the effects of vigorous physical activity in elderly people and/or in middle-age cancer therapy-treated patients. As it is a narrative review article, I believe it is not necessary to create a discussion topic. I would replace the discussion with the topic, taking advantage of the information contained therein to enrich the heart of the study, that is, high-intensity exercises, the elderly and cancer.

Concluding remarks

10) The conclusion contains several pieces of information that need to be further detailed in the topics above, as already suggested. For example, it should be addressed that light and moderate intensity exercise brings health benefits, and not just a 30-minute walk. The text also did not detail lifestyle habits such as alcohol consumption and smoking. If you want to keep this information in the conclusion, please work on it in more detail in the text.

Author Response

Comment 1) Cite the reference used to prepare the paragraph located between lines 79 and 83. Also remove the underlined information in this paragraph.

Answer. I accept this justified comment. Underlined text was removed and two references are now cited to support that statement - Van der Zander et al. FEBS J 288 2020 and Stanfield A et al. Exp Opin Pharmacother 23, 2022 - which were added to the Reference section.

Comment 2) In the caption for figure 1, state where the information was extracted for its creation, as well as explaining better in the text all types of treatment and chronic side effects on the affected organs and/or regions of the body.

Answer. I accept some of the proposed changes. I could not find anymore where the center picture came from. I therefore replace it with a comparable image from Freepik. It is not clearly stated in the legend section. For the remaining comment, I consider that it is beyond the scope of this short review to expend on the detailed drugs, treatments, and their long list of long-term complications. I instead now refer the readers to an appropriate reference for further details (Thong MSY et al. 2025).

Comment 3) In line 100, it is mentioned that the age factor is related to the increase in heart rate variability, and is subsequently pointed out as something bad, however, the literature indicates that the greater the heart rate variability, the better the health. Please check this statement and adjust.

Answer. I agree. It is a mistake. The sentence has been corrected accordingly (i.e., HRV is indeed decreased in elderly).

Comment 4) Regarding figure 2, as mentioned in figure 1, please cite the authors who were used as support for the creation of this image. In addition, detail in the text the possible physiological mechanisms that lead to the outcomes of the changes proposed in figure 2.

Answer. I agree. The reference to Freepik and the author is now specified in the legend. Regarding the details mechanisms, the readers are now referred to a proper review article (Preston and Biddell, 2021).

Comment 5) Cite the reference used to create the first paragraph of this topic (between lines 123 and 133).

Answer. Thank. I agree. I now include this link from the CDC. https://www.cdc.gov/physical-activity-basics/benefits/index.html

Comment 6) In topic 4, the first paragraph provides information about the importance of physical activity and the second paragraph mentions the harmful effects of physical inactivity. However, this information is very superficial and requires more details, such as mentioning the intensity of the effort (light or moderate), immunological and behavioral changes, body composition and recommendations from the World Health Organization and the American College of Sports Medicine, for example. Here are some supporting articles to strengthen this topic: https://doi.org/10.1186/s13195-020-00597-3; https://doi.org/10.1139/apnm-2020-0467; https://doi.org/10.1590/1517-869220182405185533; https://doi.org/10.3390/medsci8010011; https://doi.org/10.1155/2018/5230971.

Answer. I agree that additional details exist and could be provided. However, given that it is a short review and that this knowledge is generally well-known by experts and even non-experts, no other details and references will be provided here.

Comment 7) The assertion between lines 194 and 198 needs to be referenced.

Answer. All right. Here's the reference added to the manuscript: Tschopp and Brunner 2017

Comment 8) In figure 3, it is important to mention in front of the topics which author is related to each statement, this makes it easier for the reader to search for the information, if necessary.

Answer. I disagree. This would make the figure overloaded with unnecessary information given that, again, it is only a short review.

Commen 9) Before the discussion and after topic 5, I believe it is necessary to create a topic focused on the central theme of the review, which would specifically address the objective of the study, that is, the effects of vigorous physical activity in elderly people and/or in middle-age cancer therapy-treated patients. As it is a narrative review article, I believe it is not necessary to create a discussion topic. I would replace the discussion with the topic, taking advantage of the information contained therein to enrich the heart of the study, that is, high-intensity exercises, the elderly and cancer.

Answer. This comment is well-taken. However, no change will be made.

Comment 10) The conclusion contains several pieces of information that need to be further detailed in the topics above, as already suggested. For example, it should be addressed that light and moderate intensity exercise brings health benefits, and not just a 30-minute walk. The text also did not detail lifestyle habits such as alcohol consumption and smoking. If you want to keep this information in the conclusion, please work on it in more detail in the text.

Answer. I agree with the first part. I added :  '... or light and moderate intensity exercise...'

Round 2

Reviewer 2 Report

Comments and Suggestions for Authors

Dear Editor,

After verifying that the author has made improvements to the manuscript, my suggestion is that the study be accepted for publication.